SOFTWARE

# PyRates—A code-generation tool for modeling dynamical systems in biology and beyond

**Richard Gast**[1,2]*, **Thomas R. Knösche**[3], **Ann Kennedy**[1,2]

**1** Department of Neuroscience, Feinberg School of Medicine, Northwestern University, Chicago, Illinois, United States of America, **2** Aligning Science Across Parkinson's (ASAP) Collaborative Research Network, Chevy Chase, Maryland, United States of America, **3** Brain Networks Group, Max Planck Institute for Human Cognitive and Brain Sciences, Leipzig, Germany

* richard.gast@northwestern.edu

**Data Availability Statement:** Scripts to reproduce figures 2-4 are available at https://github.com/pyrates-neuroscience/use_examples/.

**Funding:** The study is funded by the joint efforts of The Michael J. Fox Foundation for Parkinson's

## Abstract

The mathematical study of real-world dynamical systems relies on models composed of differential equations. Numerical methods for solving and analyzing differential equation systems are essential when complex biological problems have to be studied, such as the spreading of a virus, the evolution of competing species in an ecosystem, or the dynamics of neurons in the brain.

Here we present *PyRates*, a Python-based software for modeling and analyzing differential equation systems via numerical methods. *PyRates* is specifically designed to account for the inherent complexity of biological systems. It provides a new language for defining models that mirrors the modular organization of real-world dynamical systems and thus simplifies the implementation of complex networks of interacting dynamic entities. Furthermore, *PyRates* provides extensive support for the various forms of interaction delays that can be observed in biological systems.

The core of *PyRates* is a versatile code-generation system that translates user-defined models into "backend" implementations in various languages, including *Python*, *Fortran*, *Matlab*, and *Julia*. This allows users to apply a wide range of analysis methods for dynamical systems, eliminating the need for manual translation between code bases.

*PyRates* may also be used as a model definition interface for the creation of custom dynamical systems tools. To demonstrate this, we developed two extensions of *PyRates* for common analyses of dynamic models of biological systems: *PyCoBi* for bifurcation analysis and *RectiPy* for parameter fitting. We demonstrate in a series of example models how *PyRates* can be used in combination with *PyCoBi* and *RectiPy* for model analysis and fitting. Together, these tools offer a versatile framework for applying computational modeling and numerical analysis methods to dynamical systems in biology and beyond.

Research (MJFF) and the Aligning Science Across Parkinson's (ASAP) initiative. The salary of R.G. was paid via the grant ASAP-020551 which A.K. received and which MJFF administers on behalf of ASAP and itself. The funders had no role in study design, data collection and analysis, decision to publish, or preparation of the manuscript.

**Competing interests:** The authors have declared that no competing interests exist.

## Author summary

We present *PyRates*, a code-generation tool for dynamical systems modeling applied to biological systems. Together with its extensions *PyCoBi* and *RectiPy*, *PyRates* provides a framework for modeling and analyzing complex biological systems via methods such as parameter sweeps, bifurcation analysis, and model fitting. We highlight the main features of this framework, with an emphasis on new features that have been introduced since the initial publication of the software, such as the extensive code generation capacities and widespread support for delay-coupled systems. Using a collection of mathematical models taken from various fields of biology, we demonstrate how *PyRates* enables analysis of the behavior of complex nonlinear systems using a diverse suite of tools. This includes examples where we use *PyRates* to interface a bifurcation analysis tool written in *Fortran*, to optimize model parameters via gradient descent in *PyTorch*, and to serve as a model definition interface for new dynamical systems analysis tools.

## Introduction

Scientists have been using differential equation systems to study real-world dynamical systems since the formulation of classical mechanics by Newton [1–4]. Disciplines as diverse as physics, biology, neuroscience, and earth sciences have applied differential equation systems to model phenomena such as fluid dynamics, population growth, neural synchronization, and climate change. While some simple differential equation systems have analytical solutions, most real-world systems are too complex to study analytically [3]. Hence, numerical methods are critical to gain a scientific understanding of differential equation systems [5, 6]. Numerical methods can find solutions to complex problems such as the prediction of weather changes [7], the conditions for an ecosystem to approach a stable state [8], or the optimal application of electrical stimulation for treating a neurological disorder [9]. Solving these problems using numerical methods can involve the integration of differential equation systems with thousands of state variables, the application of automated parameter optimization algorithms in high-dimensional parameter spaces, or the automated detection of stable solutions of differential equation systems.

The research community has developed many software packages that efficiently implement the most widely used numerical analyses for dynamical systems (see Table 1 for examples). However, there is no standardization in how dynamical systems models must be formulated or how analysis models are implemented across different packages. Additionally, different packages differ in their degree and style of software documentation, versioning, and automated testing. These idiosyncrasies impede the adoption, reproducibility, shareability, and transparency of numerical dynamical system analysis results [10–12]. Here, we present *PyRates*, an

**Table 1. Exemplary list of dynamical systems analysis software packages that are supported by one of the backends available in *PyRates*.**

| Name | Description | Backend |
|---|---|---|
| *DifferentialEquations.jl* [14] | Toolbox for numerical analysis of various types of differential equation systems | *Julia* |
| *BlackBoxOptim.jl* [15] | Toolbox for model-independent parameter optimization | *Julia* |
| *Auto-07p* [16] | Toolbox for numerical parameter continuation and bifurcation analysis of differential equation systems | *Fortran* |
| *SciPy* [17] | Toolbox that includes methods for differential equation integration and parameter optimization | *NumPy* |
| *PyTorch* [18] | A machine learning library that includes methods for gradient-based parameter optimization | *PyTorch* |
| *pygpc* [19] | Model-independent sensitivity and uncertainty analysis toolbox | *NumPy* |
| *DDE-BIFTOOL* [20] | Toolbox for numerical parameter continuation and bifurcation analysis of delayed differential equation systems | *Matlab* |

open-source *Python* toolbox for dynamical systems modeling that addresses these problems by allowing users to access a wide variety of dynamical systems tools (e.g. the ones listed in Table 1) from a single model implementation.

## Main features of PyRates

While *PyRates* was previously developed as a toolbox for neural network simulations [13], it has since evolved into a general framework for modeling and analyzing dynamical systems. We have made significant updates, including the addition of new backends, substantial expansion of the software's code generation capabilities, support for delay differential equation systems, and the introduction of new interfaces that enable the use of *PyRates* as a model definition interface for other tools. In its current form, *PyRates* extensively supports the demands for modeling complex biological systems. Most importantly, *PyRates* provides a variety of methods to model delays in the interaction between dynamical processes, such as the synaptic transmission delay between interacting neurons, or the delay until the reduction in the population size of a species affects the stability of an ecosystem. The purpose of this paper is to demonstrate these capabilities, and establish *PyRates* as a code generation tool for dynamical systems methods applied to biological and physical systems.

*PyRates* provides a flexible model definition language, which is parsed by the library's code-generation tools into output code that can be run in various third-party software packages or "backends". The model definition language enables users to define simple mathematical operators (differential equation systems) and connect them hierarchically to form networks of interacting elements. Models defined via *PyRates* can be translated into other programming languages by *PyRates*'s code generation system, and this auto-generated code then run independently in any supported backend (see Table 1 for examples). *PyRates* thus provides easy access to the diverse dynamical systems analysis tools that these backends provide, without requiring the user to re-implement dynamical systems models in each new language. For example, the same model definition can be used to perform parameter optimization via the *Julia* toolbox *BlackBoxOptim.jl* [15] and bifurcation analysis via the *Fortran*-based software *Auto-07p* [16]. Thus, *PyRates* offers (i) a simplified process for implementing dynamical system models with minimal potential for errors, (ii) a transparent model definition language that simplifies sharing and reproducing model analyses, and (iii) access to a wide range of dynamical system analysis packages through its code generation approach.

In the following sections, we first compare *PyRates* to other, related dynamical systems modeling software. We then present the software structure of *PyRates* in detail, noting novel features that have been added since our previous manuscript [13]. This is followed by use cases that demonstrate the main features of *PyRates* using a number of well-known dynamical system models that previously have been applied to model biological systems. Finally, we discuss the potential of the software to advance the application of dynamical systems methods to biological questions.

## PyRates in comparison to other dynamical systems tools

As shown in the Results section, *PyRates* supports numerical integration of differential equation systems and parallelized parameter sweeps. While this feature is useful for model validation and small dynamical system analyses, it is not the main purpose of the software. Other tools such as *DifferentialEquations.jl* for *Julia* [14], *SciPy* for *Python* [17], or *XPPAUT* for *Matlab* [21] offer a wide range of numerical differential equation solvers. The main advantage of *PyRates* is that it allows users to interface these tools from a single model definition, giving them the flexibility to choose the best solver for their purposes.

Regarding the model definition, *PyRates* is the framework with the most extensive support for modeling delayed interactions between dynamical processes. Each connection between variables in a dynamical system can be given either a fixed delay, or a distributed delay. Distributed delays are automatically translated into sets of coupled differential equations, thus making them compatible with any ordinary differential equation solver. Finally, *PyRates* provides support for defining delayed differential equations, including interfaces to analysis tools for delayed differential equations such as *DDE-BIFTOOL* [20].

*PyRates*'s code-generation approach sets it apart from dynamical system modeling frameworks such as COMSOL MultiPhysics [22], *PyDS* [23], *PySD* [24], *Simupy* [25], *The Virtual Brain* [26, 27], the *Brain Dynamics Toolbox* [28], or the Brain Modeling Toolkit [29], which provide a range of dynamical system analysis methods within a single framework. These tools can be useful for minimizing implementation errors, and for users that want a single tool with a set of analysis and visualization options. However, if a specific analysis method or algorithm is not provided, these tools lack the flexibility to interface with third-party software. In contrast, *PyRates*'s code-generation approach allows users to choose the best algorithms and implementations for each step in a dynamical system analysis pipeline. For example, given a single model definition, *PyRates* can export one piece of code to use `scipy.optimize` to fit your model to data, another to use *DDE-BIFTOOL* for bifurcation analysis around the optimized parameter set, and finally a third to generate time series in different parameter regimes via *DifferentialEquations.jl*.

This code-generation framework makes *PyRates* similar to tools such as *Brian* [30], *ANNarchy* [31], *RateML* [32], *NESTML* [33], *NeuroML* [34], or *CellML* [35]. All of these tools generate code from user-defined model equations and are designed for numerical integration of neurodynamic models. Their use of code generation allows users to design a custom model via the software frontend, and obtain optimized code for backend implementation that is efficient on specific hardware. However, they each only generate code for a specific third-party backend (such as C or *Python*), and the generated code is not directly accessible to the user. *PyRates*, on the other hand, provides inherent access to the code generated for its different backends, while still offering run-time optimization options such as vectorizing the model equations or using function decorators like *Numba* [36] (see the gallery example on run-time optimization at https://pyrates.readthedocs.io/en/latest/). The user can easily manipulate *PyRates*-generated code, for example to embed it into other scripts, thus maintaining full control even after the model is translated into a specific backend. This is an advantage over other string-based code generation methods, as the generated code can be easily observed and analyzed. Therefore, *PyRates* is attractive to experts and scholars in dynamical system modeling. It provides the flexibility to implement complex models and use expert-level analysis tools, while also offering full control over the model equations, allowing scholars to examine and adjust the output of *PyRates* and gain a deeper understanding of the models and analysis techniques.

In summary, *PyRates* is more than just a differential equation solver. It is a dynamical system modeling framework that offers a range of differential equation solving options, but mostly stands out for (i) a simple, yet powerful model definition language, (ii) its extensive support for modeling biological interaction delays, and (iii) translating these models into equation files for interfacing with other dynamical system tools.

## Design and implementation

*PyRates* consists of a frontend and a backend. The frontend provides a user-friendly interface for model definition, numerical simulations, and code generation, while the backend allows

efficient evaluation of the model equations using a number of powerful programming languages and toolboxes. See Fig 1 for a visualization of this structure.

## The PyRates frontend

*PyRates* allows the implementation of dynamical system models of the form

$$\dot{\mathbf{y}} = \mathbf{F}(\mathbf{y}(t), \theta, t, \mathbf{y}(t - \tau_1), \dots, \mathbf{y}(t - \tau_n)), \tag{1}$$

with $N$-dimensional state-vector $\mathbf{y}$ and $N$-dimensional vector-field $\mathbf{F}$. This vector field can depend on the current state of the system $\mathbf{y}(t)$ as well as previous states of the system $\mathbf{y}(t - \tau_i)$ $\forall\, i \in 1, \dots, n$, a parameter vector $\theta$ and time $t$. Thus, *PyRates* supports the implementation of autonomous and non-autonomous dynamical systems, and allows for the use of ordinary and delayed first-order differential equation systems. For more information on the mathematical framework and syntax supported by *PyRates*, see https://pyrates.readthedocs.io/en/latest/math_syntax.html.

Dynamical systems models that follow Eq (1) are implemented in *PyRates* via a hierarchy of template classes (see Fig 1A). In the following, we will provide a brief demonstration of how to implement a model via template classes, using the example of the Lotka-Volterra equations. For a more detailed introduction to the *PyRates* model definition language, see our online documentation at https://pyrates.readthedocs.io/en/latest.

The Lotka-Volterra equations are a classic model of the dynamics of interacting predator and prey populations in an ecological system [37] where the dynamics of the population density $x_i$ of each species is given by

$$\dot{x}_i = x(\alpha_i + \sum_{j=1}^{N} \beta_j x_j), \tag{2}$$

with growth rates $\alpha$ and coupling constants $\beta$. The following Python code implements Eq (2) via an **operator template** in *PyRates*:

```
1 from pyrates import OperatorTemplate
2
3 op = OperatorTemplate (
4     name="lv",
5     equations="x' = x*(alpha + x_in)",
6     variables={"x": "output(0.5)", "alpha": 0.5,
7                "x_in": "input(0.0)"}
8 )
```

**Listing 1**. Lotka-Volterra operator template definition.

The `OperatorTemplate` is the basic functional unit of *PyRates* and is composed of a set of equations (`eq` can be either a single equation or a list of equations) and its associated input, output, and intrinsic variables. In the context of the Lotka-Volterra equations, the output is the density of the population under consideration and the input is the population density of other species that interact with the species under consideration. For each time-dependent variable, the initial condition can optionally be provided in parentheses, as in "x": "output(0. 5")`, which declares x as an output variable of the operator with initial state $x(t_0) = 0.5$.

Once the operator templates for a model are defined, they are organized into nodes and edges, where nodes represent the atomic units of the dynamical system, and edges represent coupling functions between these units (see Fig 1A). For this simple example, we need only the

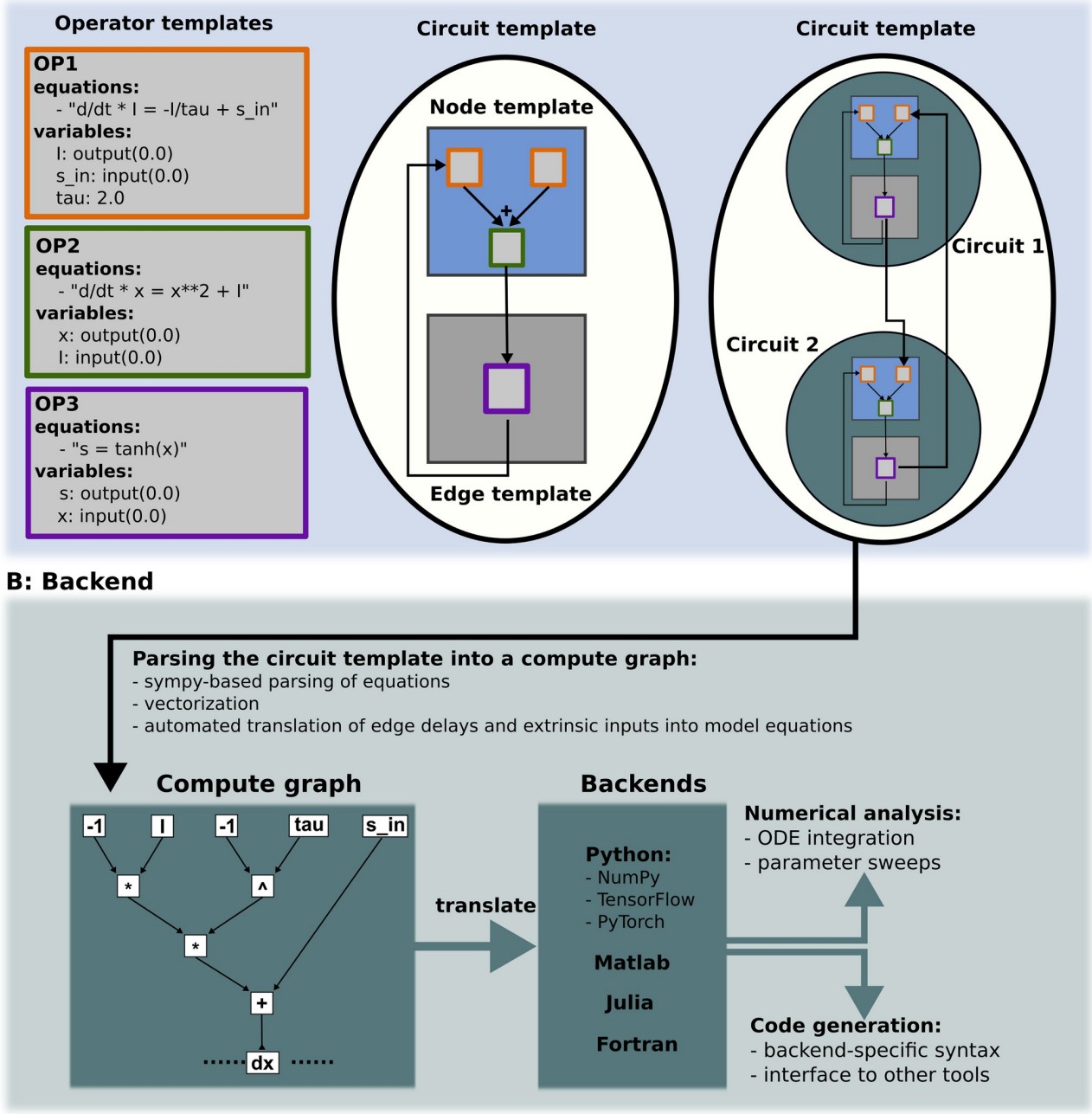

**Fig 1. *PyRates* software structure. (A)** Depiction of the user interface: *PyRates* models are implemented via different templates that can be defined via a *YAML* or *Python* interface. `OperatorTemplate` instances are used to define equations and variables and serve as basic building blocks for `NodeTemplate` and `EdgeTemplate` instances. The latter can be used to define `CircuitTemplate` instances which are used to represent the final models in *PyRates*. `CircuitTemplate` instances can also be incorporated in higher-level `CircuitTemplate` instances to allow for complex hierarchies, as depicted by the coupling of *circuit 1* and *circuit 2* within a `CircuitTemplate`. **(B)** Structure of the backend: Each model is translated into a compute graph, which in turn is parsed into a backend-specific model implementation. The latter can be used for code generation and numerical analyses.

`op` operator that we defined above. To set up the Lotka-Volterra system, we instantiate a set of nodes based on this operator, and relate them through a set of edges:

```
1  from pyrates import NodeTemplate, CircuitTemplate
2
3  # set up nodes for a predator and a prey population
4  alphas = {"predator": -1.0, "prey": 0.7}
5  nodes = {key: NodeTemplate(name="population", operators={op: {"alpha":
   alphas[key]}}) for key in ["predator", "prey"]}
6
7  # connect predator and prey population in a circuit
8  model = CircuitTemplate (
9      name="network",
10     nodes=nodes,
11     edges=[
12         ("predator/lv/x", "prey/lv/x_in", None,
13          {"weight": -1.3}),
14         ("prey/lv/x", "predator/lv/x_in", None,
15          {"weight": 1.0, "delay": 0.5})
16          ]
17       )
```

**Listing 2**. Definition of a network of interacting species via node and circuit templates.

First, we created `NodeTemplate` instances for two species, namely a predator and a prey population, each governed by the population growth rate operator that we previously defined. By providing a dictionary to the keyword argument `operators` of `NodeTemplate`, we set the species-specific growth rates $\alpha$ to values distinct from the default value of $\alpha = 0.5$ set in the operator template.

We next integrated these nodes into a `CircuitTemplate` and connected them via simple linear edges. Each edge is articulated using a tuple with four elements: the source variable, the target variable, an optional `EdgeTemplate` (not used in this example), and a dictionary holding edge attributes. When specifying a variable, such as the source variable of an edge, the following syntax should be used: "`<node>/<operator>/<variable>`". This is a unique pointer that identifies a variable in terms of the target *node* in the network, the designated *operator* on that node, and the specific *variable* of that operator. The most important edge attribute is "`weight`", which determines the edge's projection strength. In this example, we also added the "`delay`": `0.5` attribute to the edge connecting the prey to the predator population, thereby rendering the system a delay-coupled system. This feature can effectively simulate temporal delays in the species interaction [38, 39].

The showcased example demonstrates nodes defined with a single operator and straightforward linear projections for edges, which doesn't necessitate an operator. For more complex models, multiple operators can be combined to represent the underlying equations governing atomic units (nodes) and their interconnections (edges). The primary advantage of this template-based model definition lies in the reusability of each template across models, capitalizing on the commonality of mathematical models that describe many biological processes. This is shown in Fig 1A, where three operator templates are used multiple times in the nodes and edges in the circuit template. Likewise, the nodes (edges) that share a structure in Fig 1A only require a single node (edge) template for their definition. Finally, as seen in Fig 1A, the template-based model definition interface allows for the use of circuit templates to define higher-level circuits, enabling the creation of models of complex, hierarchically structured dynamical

system. This is particularly useful to examine how different hierarchical levels of complex dynamical systems interact with each other, and under which conditions lower-level dynamics may be neglected for the benefit of reduced model complexity and thus improved simulation speed. For further documentation of the template user interface and the various modeling option it provides, see https://pyrates.readthedocs.io/en/latest/template_specification.html.

## The PyRates backend

Fig 1B illustrates the working principles of the *PyRates* backend. Whenever a model template is used for simulations or code generation, the model is first translated into a compute graph. This is done using the equation parsing functionalities of *SymPy*, a well-known *Python* library for symbolic mathematics [40]. The resulting graph represents all variables and the mathematical operations connecting them, creating a flow chart from the differential equation system input to its output, i.e. the vector field of the model. Following its construction, the compute graph is translated into a backend-specific function for the evaluation of the vector field. This function can be used directly for numerical simulations of the system dynamics, or it can be written to a file, with the syntax and file type depending on the chosen backend. Currently, the following backends are available in *PyRates*:

- *NumPy* [41],

- *TensorFlow* [42],

- *PyTorch* [18],

- *Fortran 90* [43],

- *Julia* [44],

- *Matlab* [45].

Due to the modular structure and open-source nature of *PyRates*, additional backends can be added with relatively little effort. Generated function files can be used to interface other tools such as the ones listed in Table 1, or numerical integration of the model equations or parameter sweeps can directly be performed in *PyRates*. In that case, *PyRates* will automatically use the generated function file.

As an example, we generate the run function for the predator-prey model we defined in the previous section:

```
1
2  model.get_run_func (
3      "predator_prey", backend="matlab",
4      step_size=1e-3, adaptive=True
5  )
```

**Listing 3**. Numerical simulation of the Lottka-Volterra equations.

During the call of the `CircuitTemplate.get_run_func` method, *PyRates* automatically translates the model equations into a compute graph and generates a function file from the compute graph in the language of the specified backend (*Matlab*).

The `step_size` indicates at which step size the model equations are going to be integrated and the `adaptive` flag permits integration with an adaptive step size. In the latter case, `step_size` specifies the initial integration step size and the step size at which extrinsic inputs to the model would have to be defined. When `adaptive=True`, the generated function file will contain a set of delayed differential equations that could for example be used for

bifurcation analysis of the delay-coupled Lotka-Volterra equations via the *Matlab* software *DDE-BIFTOOL* [20]. If `adaptive=False`, *PyRates* would add an intrinsic buffer to the model equations which implements the interaction delays but requires a fixed integration step size. The *Matlab* file generated by the above code is provided in the supporting information (S2 File), and a script containing the Lotka-Volterra model definition as well as a simulation of the model dynamics is available at https://www.github.com/pyrates-neuroscience/use_examples.

## PyRates as a model definition interface

*PyRates* can also be used as a model definition interface for more specialized dynamical systems tools. Tools that extend *PyRates* can take advantage of its template-based, hierarchical model definition system, and use *PyRates*'s code generation capacities to translate model definitions for a target backend. Here, we present two *Python* tools that we developed using *PyRates* as their model definition interface: *PyCoBi*, for parameter continuation and bifurcation analysis, and *RectiPy* for recurrent neural network modeling. Both tools are part of the collection of open-source software provided with *PyRates* and are freely available at https://github.com/pyrates-neuroscience.

**PyCoBi.** This package provides specialized support for parameter continuation and bifurcation analysis, two common numerical computing tasks in the characterization of dynamical systems. *PyCoBi* is based on the *Fortran* software *Auto-07p*, one of the most popular and powerful tools for parameter continuations. By leveraging the code generation functionality of *PyRates*, *PyCoBi* provides a modern user interface to *Auto-07p* that does not require any *Fortran* coding (although users can also use *PyCoBi* on existing Fortran files.) We demonstrate the functionality of *PyCoBi* in the following section, where we use *PyRates* to generate *Fortran* files for a dynamical system and use those files to perform bifurcation analysis via *PyCoBi*.

**RectiPy.** This package extends *PyRates* with custom methods for recurrent neural network optimization and simulation. *RectiPy* uses *PyRates* both to define networks of recurrent rate or spiking neurons and to translate those networks into a *PyTorch* graph. The *PyTorch* graph represents the model equations, and is called at run-time to evaluate the right-hand side of Eq (1). It is thus a *PyTorch*-specific version of the compute graph created in *PyRates* via *SymPy*, and provides access to the high-level routines for gradient-based parameter optimization and numerical integration of the network equations available in *PyTorch*. We demonstrate the functionalities of *RectiPy* and how it integrates *PyRates* as a user interface in the following section.

## Results

In this section, we demonstrate different stages of the *PyRates* workflow using differential equation systems that have previously been applied to model biological systems. We show how different dynamical system analysis methods can be applied to these models via *PyRates*, and demonstrate the flexibility that *PyRates* offers in analyzing dynamical system model dynamics and parameter dependencies.

The dynamical system models used in the examples below come pre-implemented with *PyRates* and are explained in detail in our online documentation at https://pyrates.readthedocs.io/en/latest/. Furthermore, we provide a *YAML* file that contains the definitions of all models used below in the supporting information (S1 File). Scripts to reproduce the results and figures of each of our use examples at https://www.github.com/pyrates-neuroscience/use_examples. Both these scripts and the code snippets below were developed and tested with the following software versions: *PyRates 1.0.4* [46], *PyCoBi 0.8.5* [47], and

*RectiPy 0.12.0* [48]. You can download the full source code for these versions by following the DOIs provided in the references.

## Using *PyRates* for numerical simulations and parameter sweeps

In this example, we demonstrate how *PyRates* can be used to perform numerical simulations and parameter sweeps. We study a Van der Pol oscillator, an oscillator model with non-linear damping which has been considered as a phenomenological model of the heartbeat [49]. Driving the Van der Pol oscillator with periodic input from a simple Kuramoto oscillator, a model which has been applied to oscillatory processes as diverse as the synchronization of spiking neurons [50] or the interactions of social agents [51], we examine its entrainment to the Kuramoto oscillator frequency as a function of the input strength and frequency. In the context of cardiac modeling, this analysis would reveal which configurations of a periodic electrical stimulation would speed up or slow down the heartbeat. We perform this analysis using a *PyRates* function that executes multiple, vectorized numerical integrations of the differential equation system (3–5), one for each parametrization of interest. The equations of the system are:

$$\dot{x} = z, \tag{3}$$

$$\dot{z} = \mu z(1 - x^2) - x - J sin(2\pi\theta), \tag{4}$$

$$\dot{\theta} = \omega. \tag{5}$$

The state variables of the differential equation system (3–5) are the Van der Pol oscillator state variables $x$ and $z$ and the Kuramoto oscillator phase $\theta$, and the system parameters are given by the damping constant $\mu$, the input strength $J$, and the intrinsic frequency of the Kuramoto oscillator $\omega$.

Equations for both the Van der Pol and Kuramoto oscillators are pre-implemented in *PyRates*. For comprehensive reviews of the properties of these oscillators, see [52, 53]. The following code uses the `NodeTemplate` class to load the definitions of the Van der Pol oscillator and Kuramoto oscillator, then uses the `CircuitTemplate` class to define the network of nodes and edges that make up the dynamical system given by Eqs (3)–(5):

```
1  from PyRates import CircuitTemplate, NodeTemplate
2
3  # define nodes
4  VPO = NodeTemplate.from_yaml (
5      "model_templates.coupled_oscillators.vanderpol.vdp_pop"
6      )
7  KO = NodeTemplate.from_yaml (
8      "model_templates.coupled_oscillators.kuramoto.sin_pop"
9      )
10
11 # define network
12 net = CircuitTemplate (
13     name="VPO_forced", nodes={'VPO': VPO, 'KO': KO},
14     edges=[('KO/sin_op/s', 'VPO/vdp_op/inp', None, {'weight':1.0})]
15     )
```

**Listing 4**. Definition of the Van der Pol oscillator model.

With the model loaded into *PyRates*, we can use numerical integration to generate time series of its dynamics for different values for *J* and *ω*. To minimize the runtime of this problem, we use the function `pyrates.grid_search`, which takes a set of multiple model parametrizations and performs the numerical integration in a single combined model by vectorizing the model equations. The code below defines a parameter sweep with 20 values of *J* and 20 values of *ω*, resulting in $N = 400$ model parametrizations.

```
1  # imports
2  import numpy as np
3  from PyRates import grid_search
4
5  # define parameter sweep
6  n_om = 20
7  n_J = 20
8  omegas = np.linspace(0.3, 0.5, num=n_om)
9  weights = np.linspace(0.0, 2.0, num=n_J)
10
11 # map sweep parameters to network parameters
12 params = {'omega': omegas, 'J': weights}
13 param_map = {'omega': {'vars': ['phase_op/omega'],
14                        'nodes': ['KO']},
15              'J': {'vars': ['weight'],
16                    'edges': [('KO/sin_op/s', 'VPO/vdp_op/inp')]}
17             }
18
19 # perform parameter sweep
20 results, res_map = grid_search(
21     circuit_template=net, param_grid=params, param_map=param_map,
22     simulation_time=T, step_size=dt, inputs=None, vectorize=True,
23     outputs={'VPO': 'VPO/vdp_op/x', 'KO': 'KO/phase_op/theta'},
24     solver='scipy', method='DOP853', clear=False,
25     permute_grid=True, cutoff=cutoff, sampling_step_size=dts
26     )
```

**Listing 5**. Parameter sweep over periodic forcing parameters in the Van der Pol oscillator model.

The `grid_search` call takes the given `circuit_template` and creates copies of it for each set of parameters in `param_grid`. It adjusts the parameters of each copy accordingly, using the information in `param_map` to locate the parameters that should be adjusted. It then places all copies of the `circuit_template` in one big model and performs the simulation, with the remaining arguments controlling the numerical integration procedure. Given a set of *N* different model parametrizations, this procedure results in an implementation of the differential equation system (3–5) where each variable in the equations is represented by a vector of length *N*. For the numerical integration, we instructed `grid_search` to use a Runge-Kutta algorithm of order 8 with automated adaptation of the integration step size, via the flag `method="DOP853"`. This solver is available in *PyRates* through the `scipy.integrate.solve_ivp` method of *SciPy* [17]. Alternative choices of numerical integration methods are available via the keyword arguments `solver` and `method` of `grid_search`. For more details on how to specify parameter sweeps via the `grid_search` function and the different

options available to adjust the behavior of the function, see https://pyrates.readthedocs.io/en/latest/auto_analysis/parameter_sweeps.html.

We use the returned values of the `grid-search` call in Listing 5 to compute the coherence between $\theta$ and $x$ for each set of $\omega$ and $J$. This results in a triangularly shaped coherence profile, also know as an Arnold tongue (see Fig 2A), which describes the characteristic entrainment behavior of a non-linear oscillator subject to a periodic driving force [54].

The larger the difference between the driving frequency and the intrinsic frequency of the non-linear oscillator (or one of its harmonics), the stronger the required amplitude of the driving signal for entraining the oscillator to the driving frequency. Our example confirms that the Van der Pol oscillator expresses this behavior. In Fig 2B, we show an example where the driving force $J$ was too small to entrain the oscillator given the substantial difference between $\omega$ and the intrinsic frequency of the oscillator. In Fig 2C, on the other hand, we show an example where the driving force $J$ was sufficiently high and the difference between $\omega$ and the intrinsic frequency of the oscillator was sufficiently low to entrain the oscillator.

Thus, the above example demonstrates how *PyRates* can be used to perform the first steps of any dynamical system analysis, numerical integration of the differential equation system and parameter sweeps. We studied the entrainment of the Van der Pol oscillator to the Kuramoto oscillator frequency as a function of the input strength and frequency. We used the *PyRates* function `pyrates.grid_search` to perform multiple, vectorized numerical integrations of the differential equation system. The results confirm previous findings on the entrainment of a non-linear oscillator and show that the numerical integration and parameter sweep functionalities of *PyRates* work as expected.

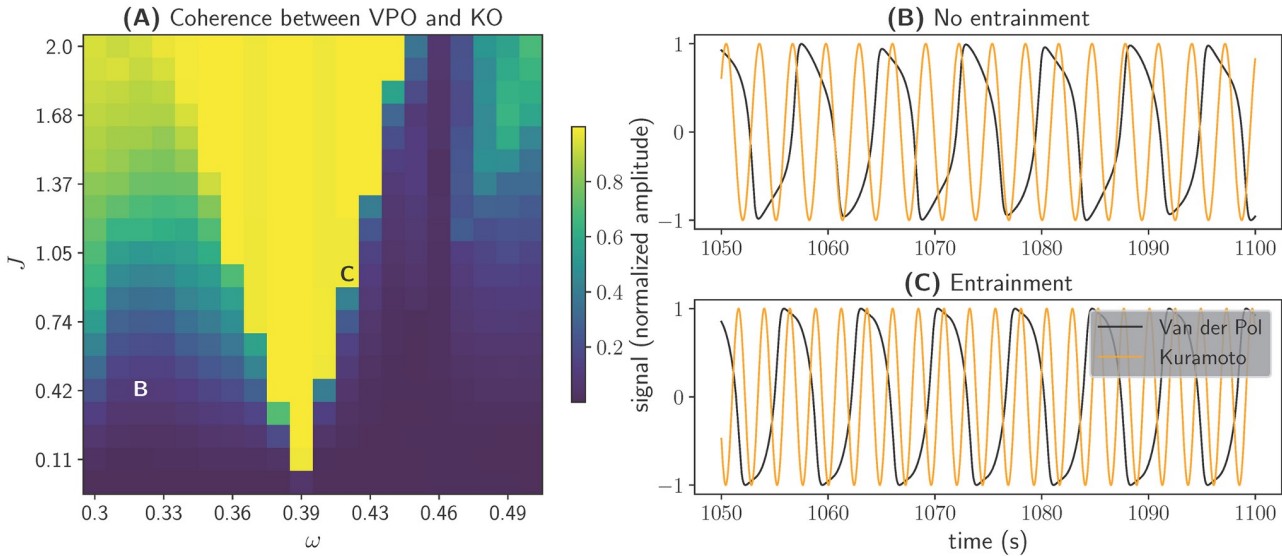

**Fig 2. Entrainment of the Van der Pol oscillator in response to periodic forcing.** (A) Coherence between the state variables $x$ of the Van der Pol oscillator (VPO) and $\theta$ of the Kuramoto oscillar (KO). For each pair of $\omega$ and $J$, we bandpass-filtered $x$ at the frequency $\omega$ and extracted the phase of the bandpass-filtered signal via the Hilbert transform. We then created a sinusoidal signal from the VPO and KO phases and used `scipy.signal.coherence` to calculate the coherence between the two sinusoids. The result is depicted as color-coding. (**B and C**) State variables $x$ (black) and $\theta$ (orange) displayed over time. (**B**) No entrainment of the VPO phase for $\omega = 0.33$ and $J = 0.5$. (**C**) Entrainment of the VPO phase to the KO phase for $\omega = 0.42$ and $J = 1.0$.

## Using PyRates for bifurcation analysis

In this example we present *PyCoBi*, one of *PyRates*'s extensions for applying numerical bifurcation analysis to a dynamical system model. Numerical bifurcation analysis is an essential tool to study qualitative changes in model dynamics caused by small variations in model parametrization [6, 55].

In neuroscience, bifurcation analysis can identify transitions between different firing regimes of a model neuron or neural population. To demonstrate this, we study the neurodynamic model described in detail in [56], which is a mean-field model of coupled quadratic integrate-and-fire (QIF) neurons with spike-frequency-adaptation:

$$\tau\dot{r} = \frac{\Delta}{\pi\tau} + 2rv, \tag{6}$$

$$\tau\dot{v} = v^2 + \bar{\eta} + I(t) - a + Jr\tau - (\pi r\tau)^2, \tag{7}$$

$$\tau_a\dot{a} = x, \tag{8}$$

$$\tau_a\dot{x} = \alpha\tau_a r - 2x - a. \tag{9}$$

The state variables of this model are $r$ and $v$, the average firing rate and membrane potential of the QIF population, and $a$ and $x$, which describe the spike-frequency-adaptation dynamics. While these equations have been derived from a network of coupled spiking neurons, they do not explicitly model spiking neurons, but rather the macroscopic dynamics of the network. Because these macroscopic equations do not contain discontinuities such as single neuron spiking events, they can be analyzed via methods from bifurcation theory. For more details on the model equations and constants, see [56].

We are interested in the effects of the adaptation strength $\alpha$ and the average neural excitability $\bar{\eta}$ on population dynamics, and would use *Auto-07p* [16] to carry out the bifurcation analysis. The goal is to reproduce the bifurcation diagrams reported in [56], where the effects of $\alpha$ and $\bar{\eta}$ on the dynamics of (6–9) have already been investigated.

*Auto-07p* requires used-supplied *Fortran* files that include the model equations and constants. As demonstrated below, *PyRates* can be used to generate these files. First, we need to load the model into *PyRates*. Since the dynamical system given by (6–9) exists as a pre-implemented model in *PyRates*, this can be done via a single function call:

```
1 from PyRates import CircuitTemplate
2 qif = CircuitTemplate.from_yaml(
3     "model_templates.neural_mass_models.qif.qif_sfa"
4     )
```

**Listing 6**. Definition of the QIF model.

After the model is loaded, it can be used to generate the input required for *Auto-07p*:

```
1 qif.get_run_func(
2     func_name='qif_run', file_name='qif_sfa',
3     step_size=1e-4, backend='fortran',
4     solver='scipy', vectorize=False,
5     float_precision='float64', auto=True
6 )
```

**Listing 7**. Auto-07p file generation via PyRates.

This method generates two files required to run *Auto-07p*: a *Fortran 90* file containing the model equations and a simple text file containing the meta parameters of *Auto-07p*. The equation file, which includes a vector field evaluation function named `func_name`, can be found in the location indicated by `file_name`; the meta parameter file, named `c.ivp`, is in the same directory. Providing the keyword argument `auto=True`, ensures that the output files are in a format compatible with *Auto-07p*.

At this point, *PyRates* has generated the meta parameters and equation files needed for parameter continuations and bifurcation analysis in *Auto-07p*. To demonstrate this, we use *PyCoBi*, which allows the calling of *Auto-07p* functions from *Python*. In Listing 8, we perform a simple numerical integration of the differential equation system over time, allowing it to converge to a steady-state solution that we can then further analyze via parameter continuations. For the example to execute without errors, provide a path to the installation directory of *Auto-07p* via `auto_dir=<path>`.

```
1 # initialize PyCoBi
2 from pycobi import ODESystem
3 qif_auto = ODESystem(working_dir=None, auto_dir=<path>,
4                      init_cont=False)
5
6 # perform numerical integration
7 t_sols, t_cont = qif_auto.run(
8     e='qif_sfa', c='ivp', name='time', DS=1e-4, DSMIN=1e-10,
9     EPSL=1e-08, EPSU=1e-08, EPSS=1e-06, DSMAX=1e-2,
10    NMX=1000, UZR={14: 5.0}, STOP={'UZ1'}
11 )
```

**Listing 8**. Numerical integration of the QIF model via *Auto-07p*.

The arguments provided to `ODESystem.run` are mostly identical to the arguments required to run *Auto-07p*, which are explained in detail in the documentation at: https://github.com/auto-07p/auto-07p/tree/master/doc. Most importantly, pointers to the generated equation and meta parameters files have been provided via the arguments `e='qif_sfa'` and `c='ivp'`, respectively.

In Fig 3B, we see that the QIF mean-field model converged to a steady-state solution within the provided integration time of the differential equation system (6–9).

Starting from this steady-state solution, we can perform parameter continuations and automated bifurcation analysis. We first continue the steady-state solution we calculated previously in the background input parameter $\bar{\eta}$. To do so, we make use of the pseudo-arclength continuation method with automated bifurcation detection, implemented in *Auto-07p* [16] and interfaced here through *PyCoBi*. This continuation method starts from the fixed point solution that the system converged to (see Fig 3B); to find a solution for a different value of the bifurcation parameter $\bar{\eta}$, a small perturbation is applied to the known solution and a new solution is found in its vicinity. The pseudo-arclength continuation method determines both the size of the perturbation in the continuation parameter and the approximate location of the new solution by making a step in the tangent space of the curve of solutions that the algorithm attempts to follow [6]. By successively perturbing the most recently found solution and searching for a new solution, the algorithm numerically approximates a curve of fixed point solutions in the bifurcation parameter space. Bifurcation points along such a curve can be detected by monitoring the local eigenvalues of the linearized vector field around the solutions. As can be seen in Fig 3A, the steady-state solution branch undergoes a number of bifurcations within the examined range of $\bar{\eta}$: Two fold bifurcations and two sub-critical Hopf bifurcations. By continuing the

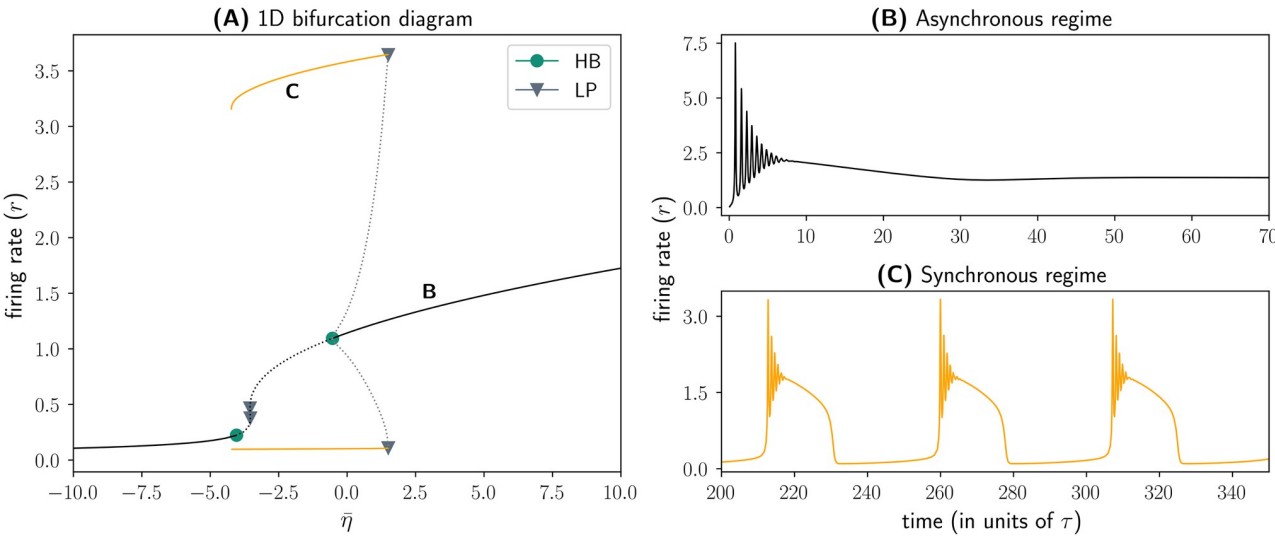

**Fig 3. Bifurcation analysis of the QIF model.** (A) Bifurcation diagram showing the solutions of Eqs (6)–(9) in the state variable *r* as a function of the parameter $\bar{\eta}$. Solid (dotted) lines represent stable (unstable) solutions. Bifurcation points are depicted as symbols along the solution branches. Green circles represent Hopf bifurcations whereas grey triangles represent fold bifurcations. (B) Convergence of the average firing rate *r* of the QIF model to a steady-state solution in the asynchronous, high-activity regime ($\bar{\eta} = 3$). (C) Convergence of the average firing rate *r* of the QIF model to a periodic solution in the synchronous, oscillatory regime ($\bar{\eta} = -2$).

unstable periodic solutions emerging from the latter, we next identified fold of limit cycle bifurcations that give rise to a regime of synchronized oscillations (see Fig 3C). The code to reproduce the bifurcation analysis results in Fig 3A can be found at https://www.github.com/pyrates-neuroscience/use_examples.

These results confirm the findings reported in [56], where a more detailed description of the QIF model's bifurcation structure is provided. Thus, we have successfully demonstrated that *PyRates* provides an interface to the parameter and bifurcation analysis software *Auto-07p*, one of the most powerful tools for studying solutions of differential equation systems and how they change with underlying system parameters.

## Parameter fitting in a delay-coupled leaky integrator model

In our final example, we demonstrate the capacity of *PyRates* as a model definition interface for other tools and show how it can support the definition of large-scale, delay-coupled dynamical systems, which are ubiquitous in biological systems due to inherently limited signal transmission speed [57, 58]. To do so, we show how *RectiPy* leverages *PyRates* as a frontend and allows *PyRates* models to be optimized using any *PyTorch* parameter optimization routine. In the example below, we use *RectiPy* for gradient-based parameter optimization in a recurrent neural network model with delay coupling that is implemented in *PyRates*.

**Building the network with *RectiPy*.** We use a set of *N* leaky integrators with non-linear delay-coupling as an exemplary model. Similar models have been applied in various biological domains such as epidemiology, population dynamics, or neuroscience [38, 39]. In neuroscience, it has been used as a phenomenological model of neurons coupled via synapses with synaptic transmission delays [59]. The model dynamics are determined by the following evolution

equations:

$$\dot{u}_i = -\frac{u_i}{\tau} + I_{ext}(t) + k\sum_{j=1}^{N} J_{ij} \tanh(\Gamma_{ij} * u_j), \tag{10}$$

$$\Gamma_{ij}(t) = \frac{a_{ij}^{b_{ij}} t^{b_{ij}-1} e^{a_{ij}t}}{(b_{ij} - 1)!}, \tag{11}$$

where $\tau$ is a global decay time constant, $k$ is a global coupling constant, $J_{ij}$ are connection-specific coupling strengths, and $I_{ext}$ is a variable that allows for extrinsic forcing. The term $\Gamma_{ij} * u_j$ is a convolution of the rate $u_j$ with the gamma kernel given by Eq (11). This type of gamma-kernel convolution is a popular model for delay-coupled systems with distributed delays, such as neural populations interacting via axons with different lengths [60] or viral spread with different incubation delays for the cell-to-cell transmission [61].

The following code implements a network of $N = 5$ coupled leaky integrators using *RectiPy*'s `Network` class, with random coupling weights and gamma kernel parameters as given by Eqs (10) and (11).

```
1  import numpy as np
2  from rectipy import Network
3
4  # network parameters
5  node = "neuron_model_templates.rate_neurons.leaky_integrator.tanh_pop"
6  N = 5
7  J = np.random.uniform(low=-1.0, high=1.0, size=(N, N))
8  D = np.random.choice([1.0, 3.0], size=(N, N))
9  S = D*0.3
10 dt = 1e-3
11
12 # initialize network
13 net = Network(dt=dt, device="cpu")
14
15 # add a recurrently coupled population of leaky integrators to the
       network
16 net.add_diffeq_node(
17     "tanh", node=node, weights=J,
18     edge_attr={'delay': D, 'spread': S},
19     source_var="tanh_op/r", target_var="li_op/r_in",
20     input_var="li_op/I_ext", output_var="li_op/u"
21     )
```

**Listing 9**. Initialization of the delay-coupled leaky integrator model in *RectiPy*.

As shown in line 16 of Listing 9, `rectipy.Network.add_diffeq_node` provides an interface for adding a *PyRates* model as a node to a `rectipy.Network` instance. `Network.add_diffeq_node` first uses `NodeTemplate.from_yaml(node)` to set up the governing equations of each network node. It then uses the connectivity weights provided via the `weights` keyword argument together with all additional edge attributes (`edge_attr`) to create a `CircuitTemplate`, and fill it with $N^2$ edges. This is implemented by a call to

`pyrates.CircuitTemplate.add_edges_from_matrix`, a method that adds edges of the following form to the network:

```
1 edge = ("<pi>/tanh_op/m", "<pj>/li_op/m_in", None,
2         {"weight": C_ij, "delay": D_ij, "spread": S_ij}
3         )
```

**Listing 10**. Definition of an edge in *PyRates*.

Here, $D_{ij}$ and $S_{ij}$ refer to the mean and variance of the gamma kernel $\Gamma_{ij}$ and are related to its parameters via $D_{ij} = \frac{a_{ij}}{b_{ij}}$ and $S_{ij} = \frac{a_{ij}}{b_{ij}^2}$. Each edge definition that includes both the "delay" and the "spread" keyword is automatically translated into a gamma kernel convolution of the source variable by *PyRates*. *PyRates* implements the convolution operation as a set of coupled differential equations that it adds to the model, using the 'linear chain trick' [61].

All string-based keyword arguments provided in lines 17–19 of Listing 9 are pointers to model variables defined in the *YAML* template specified in line 5 of Listing 9. This ensures that the network equations generated by *PyRates* are properly integrated into the *PyTorch* graph. For example, the keyword argument `input_var="li_op/I_ext"` indicates that any input provided to the network should enter the network equations via the variable `I_ext` that is defined in the operator `li_op` of the `NodeTemplate`.

Having constructed the network, *RectiPy* uses the *PyTorch* backend of *PyRates* to generate a vector field function that can be used for simulations and parameter optimization via *PyTorch*. This way, *RectiPy* extends *PyRates* to enable quick generation of *PyTorch* compute graphs from *YAML* templates. *RectiPy* can thus provide a powerful user interface for simulating and fitting recurrent neural networks with minimal coding effort.

**Performing parameter optimization in a *RectiPy* model.** We next demonstrate the use of *RectiPy* for parameter optimization. Our goal will be to recover the values of model parameters in the `rectipy.Network` instance defined in the previous section, specifically the global time constant $\tau = 2.0$ and the global coupling constant $k = 1.0$. To do so, we'll sample the model's response to a 200*Hz* sinusoidal driving input, and then use this observed response to fit the values of $k$ and $\tau$ in a second, identical model instance in which $k$ and $\tau$ are initialized from a uniform distribution over [0.1, 10.0].

We can sample the target model's activity similarly to in *PyTorch*:

```
1 # simulation parameters
2 dt = 1e-3
3 steps = 30000
4 f = 0.2
5 beta = 0.1
6
7 # simulate target signal
8 targets = []
9 for step in range(steps):
10     I_ext = np.sin(2*np.pi*freq*step*dt) * beta
11     u = net.forward(I_ext)
12     targets.append(u)
```

**Listing 11**. Generation of the target signal for parameter optimization in *RectiPy*.

Where, as in `torch.nn`, `rectipy.Network.forward` generates the output variable $u$ from the input $I_{ext}$, using the functional relationship defined by Eqs (10) and (11). Alternatively, numerical simulation can be performed in a single line with:

```
1 obs = net.run(inputs, sampling_steps=1)
```

where `inputs` is a vector of the extrinsic input to the network at each time point. This `rectipy.Network.run` method returns an instance of `rectipy.Observer`, which provides access to all network state variables recorded during the simulation.

Next, we will fit our second network model to the observed dynamics of our target network. The code example below shows how to perform a single optimization step in *PyTorch* using a mean-squared error loss function and the resilient backpropagation algorithm [62] to calculate the gradient of the error with respect to the free parameters $\tau$ and $k$.

```
1 import torch
2
3 # loss function definition
4 loss = torch.nn.MSELoss()
5
6 # optimizer definition
7 opt = torch.optim.Rprop(net.parameters(), lr=0.01)
8
9 # calculate cumulative error over entire target signal
10 mse = torch.zeros(1)
11 for step in range(steps):
12     I_ext = np.sin(2*np.pi*f*step*dt) * beta
13     u_target = targets[step]
14     u = net.forward(I_ext)
15     mse += loss(u, u_target)
16
17 # optimization step
18 opt.zero_grad()
19 error.backward()
20 opt.step()
```

**Listing 12**. Parameter optimization step in *RectiPy*.

A target signal can be fitted by iterating over optimization steps until convergence. Alternatively, the entire optimization procedure is also available via the `rectipy.Network.fit_bptt` method:

```
1 obs = net.fit_bptt(inputs, targets, optimizer="rprop", loss="mse",
    lr=0.01)
```

The results of the parameter optimization are depicted in Fig 4. As can be seen, the optimization algorithm succeeded in finding values of the parameters $\tau$ and $k$ for which the network reproduces the target dynamics of the 5 leaky integrators.

In conclusion, we successfully used the *PyTorch* equations generated by *PyRates* to run parameter optimizations via *RectiPy*.

## Availability and future directions

In this work, we presented the dynamical systems modeling software *PyRates* for simulating and interpreting the dynamics of biological systems. We focused on updates to the software since its initial release as a neural network simulation tool [13], emphasizing the new code generation capacities, additional backends, and extended support for delay-coupled systems. In a series of detailed use-cases, we showed how *PyRates* can be used to (i) perform numerical

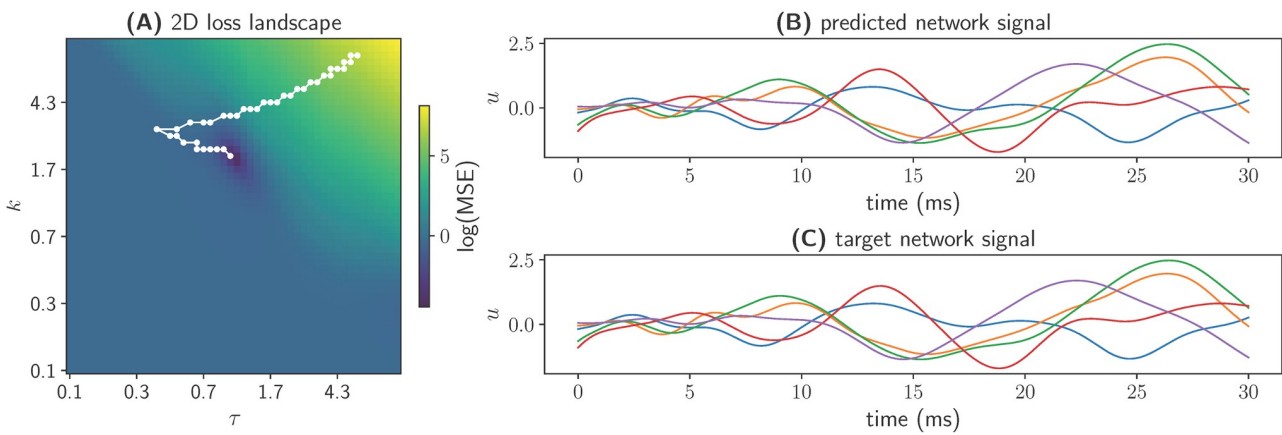

**Fig 4. Comparison of the dynamics of the target leaky integrator model and the fitted leaky integrator model. (A)** Logarithm of the mean-squared error (color-coded), depicted over the search range of the two parameters that were optimized: $\kappa$ and $\tau$. The white trace shows the steps taken by the optimizer from its initialization point to the global minimum. **(B and C)** Rate signals of all $N$ LI units over time of the fitted network and the target network, respectively.

integration of a differential equation system via `scipy.integrate.solve_ivp`, (ii) generate the Fortran files to run bifurcation analysis in *Auto-07p*, and (iii) generate the equations for a *PyTorch* compute graph to perform parameter optimization via *RectiPy*. *PyRates* thus enables the application of complex dynamical system analysis workflows to biological problems, in a manner that helps reduce implementation errors and facilitates reproducibility and shareability of mathematical models [11, 63, 64].

*PyRates* is open-source, freely available on *GitHub*, and comes with detailed documentation. Each version is released on *PyPI* and can easily be installed using the *pip* package manager. For installation instructions, see the *GitHub* repository: https://github.com/pyrates-neuroscience/PyRates. The repository also provides information on officially supported *Python* versions and the status of the extensive test library included with *PyRates*. The latter ensures that all main features and models are working as expected in the current version of *PyRates*. The structure of the software can be viewed in the API section of our documentation website: https://pyrates.readthedocs.io/en/latest/.

## Limitations

The main limitation of *PyRates* is the family of dynamical system models it supports. Currently, *PyRates* provides support for ordinary and delayed differential equation systems, and state variables of these systems can be real or complex-valued. Examples of each of these differential equation types can be found in the use example section at https://pyrates.readthedocs.io/en/latest/. *PyRates* does not currently support partial differential equations, which involve derivatives in multiple variables and can be used to model dynamical systems in continuous time and space, and stochastic differential equations, which are typically used to model inherent stochastic fluctuations of dynamical processes. Both types of differential equation systems have been widely used in dynamical system modeling [4]. In neuroscience, for instance, partial differential equations have been applied in the context of neural field models [65, 66]. A number of dynamical system analysis libraries currently supported by *PyRates* such as *SciPy* [17] or *DifferentialEquations.jl* [14] provide algorithms for the numerical integration of partial and stochastic differential equations. Thus, adding support for these types of differential equations would be a useful extension to the currently supported list of dynamical system models.

Another limitation of *PyRates* is its inability to define specific events that may occur during the numerical integration of a differential equation system. An example of such an event is the membrane potential of a neuron crossing a certain threshold and eliciting a spike, which can be modeled as a singular event in time [67]. Events like this introduce discontinuities to the differential equation system, which are not currently supported by *PyRates*. However, although *PyRates* does not support event definition in general, the *PyRates* extension *RectiPy* allows for the definition of spike conditions for the specific case of spiking neural networks. *RectiPy* provides support for numerical simulations and parameter optimization, and supports the use of both rate neurons and spiking neurons.

## Future directions

It is important to note that the limitations of *PyRates* are not inherent limitations that cannot be overcome by the software; rather, they are areas where the software has not yet been extended. Due to the highly modular structure and open-source nature of *PyRates*, such extensions can readily be implemented. For example, adding another backend to *PyRates* can be done without any changes to the frontend, whereas added support for stochastic differential equations would mostly involve changes to the frontend. Additionally, some of the limitations outlined above can be addressed by using additional software packages that extend *PyRates* with specific functionalities. We have shown here that packages that extend *PyRates* can simply be built by employing *PyRates* as a model definition interface and instructing it to generate the output files required for a specific extension. We have demonstrated that by generating *Fortran* files for *PyCoBi* and *PyTorch* files for *RectiPy*, which are software packages for bifurcation analysis and artificial neural network training, respectively.

In summary, *PyRates* already supports a large family of dynamical system systems and backends, and is designed be easily extendable in the future. This makes it a versatile dynamical system model definition language and code-generation tool that provides access to a wide variety of dynamical system analysis methods and allows for sharing models without being tied to specific programming languages or analysis tools. In the future, we aim to extend both the number of backends and the family of dynamical systems models that *PyRates* supports, thus extending its capacity as a model definition interface.

## Supporting information

**S1 File. Model definitions.** YAML file that includes the equations and model definitions of all dynamical systems models that are used in the Results section.
(YAML)

**S2 File. Matlab function file.** Matlab file that includes the forward function of the predator-prey model that is generated by Listing 3.
(M)

## Author Contributions

**Conceptualization:** Richard Gast, Thomas R. Knösche, Ann Kennedy.

**Data curation:** Richard Gast.

**Formal analysis:** Richard Gast.

**Funding acquisition:** Ann Kennedy.

**Investigation:** Richard Gast.

**Methodology:** Richard Gast.

**Project administration:** Richard Gast, Ann Kennedy.

**Resources:** Ann Kennedy.

**Software:** Richard Gast.

**Supervision:** Thomas R. Knösche, Ann Kennedy.

**Validation:** Richard Gast.

**Visualization:** Richard Gast.

**Writing – original draft:** Richard Gast, Thomas R. Knösche, Ann Kennedy.

**Writing – review & editing:** Richard Gast, Thomas R. Knösche, Ann Kennedy.

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
