## [Decision Letter · Decision Letter 0]

9 Nov 2023

Dear Mr. Gast,

Thank you very much for submitting your manuscript "PyRates - A Code-Generation Tool for Modeling Dynamical Systems in Biology and Beyond" for consideration at PLOS Computational Biology.

As with all papers reviewed by the journal, your manuscript was reviewed by members of the editorial board and by several independent reviewers. The contribution was overall very well received, yet some issues need to be addressed. In light of the reviews (below this email), we would like to invite the resubmission of a significantly-revised version that takes into account the reviewers' comments.

We cannot make any decision about publication until we have seen the revised manuscript and your response to the reviewers' comments. Your revised manuscript is also likely to be sent to reviewers for further evaluation.

Sincerely,

Daniele Marinazzo

Section Editor

PLOS Computational Biology

Daniele Marinazzo

Section Editor

PLOS Computational Biology

Reviewer's Responses to Questions

**Comments to the Authors:**

Reviewer #1: The manuscript presents a python-based software package for implementing dynamical models of ordinary- and delay-differential equations for multiple target platforms. It uses a novel modelling language to generate code for running the model on existing solvers in Python, Fortran, Matlab or Julia. This gives users the flexibility to choose the most appropriate solver for their model without altering the definition. The manuscript is well written and the ability of the software to generate code for multiple back-ends is beneficial. Although it is worth mentioning cellML (www.cellml.org) as another software package with a similar aim.

I would have liked the manuscript to begin with an example of the model definition language, say for the Van der Pol equation, before going on to describe the software structure. As it stands, the examples only demonstrate how to re-use the existing model definitions. There are no examples of how to define a new set of equations from scratch, which is where I would expect to start as a new user. Nonetheless, the current examples do provide compelling evidence of the applicability of the software to a diverse set of numerical tools across multiple programming languages.

The software makes it easy to build hierarchical models by coupling existing subsytems (operator nodes) together using operator edges. It is tempting to assume that ever larger models could be assembled using this approach. However, large hierarchical models are ultimately limited by the solver needing to accommodate the subsystem with the fastest timescale. So even though hierarchical models do have clear benefits of rapid-prototyping and code re-use, it is worth emphasizing that good model design is still needed to avoid wasting time computing the dynamics of subsystems that may have negligible impact on the global dynamics.

MINOR COMMENTS

p6. Line 144: I was confused by the example of the operator in the context of a neural population. Surely, the output (dy/dt) is the instantaneous change of the input (y). The units of synaptic current (input) and firing rate (output) in your example do not follow that rule.

p6. I felt myself wanting a little more detail about using edges to combine operator templates (nodes) using edges. Specifically, how do the outputs of some nodes become the inputs to others? An example might help.

p8. Line 244: It would be useful to see an example of the mode definition before coupling them together with the circuit template.

p11. As I understand it, the Quadratic Integrate and Fire (QIF) model involves a reset when the voltage exceeds a threshold. Yet the manuscript states (Line 580) that PyRates does not support event handling like that. Can you please clarify the apparent contradiction?

Reviewer #2: Quick summary of judgement against the criteria stated in the scope of the journal

========

Originality: the manuscript states that there are a variety of tools serving similar purpose. A unique point for the presented library is that it can serve as an interface to multiple "backends". That is, one formulates or imports the model into pyrates once and then can use several numerical engines to perform different types of mathematical analysis.

Beyond that, for intricate mathematical analysis tools such as Auto-07p the initial provision of model files (e.g. in FORTRAN) poses a significant barrier to entry for practicioners in the life sciences. The pyrates interface would make these undeservedly obscure tools accessible to a wider community.

Innovation: the tools described in the manuscript provide "glue", linking existing high-level analysis methods (bifurcation analysis, optimization, parallelized parameter studies) to easy-to-create model descriptions and a familiar interface. So, while they may not provide innovative methods in themselves, they make innovative methods accessible to a wider audience.

Importance: this would have to be measured by the level of adoption in the future, but as a developer/maintainer of mathematical bifurcation analysis tools, I am keen to have these interfaces available.

While the software discussed has the potential to support future significant biological and methodological insights and conclusions these criteria are not fully applicable to a manuscript describing a new software tool.

All demonstrations presented in the article are downloadable from an open-source repository.

In summary, I think that PLOSCOMPBIOL is a good audience for the manuscript.

========

List of concerns to be addressed in revision

========

1) The introduction (line 85ff) mentions dde-biftool as an example of a backend. Also further down support for bifurcation analysis for systems with delay is mentioned, but neither the article nor the webpage https://pyrates.readthedocs.io/en/latest have any demonstration that I could find. I would be interested to see how this works (it would require interfacing with octave or matlab). If this backend is not yet available then it should not be shown in the table or advertised in the intro as "done", but rather as something that could be supported (in the future, or if specific interest arises).

2) The section with the Frontend description uses specific technical terms without justification, eg, "operator template" or "circuit template". These are never explained even though these look like some of the key concepts behind pyrates. For example, "circuit template" appears to be a way to define a (recursive?) graph where nodes are either themselves again circuit templates or operator templates. Edges appear to be defined by tuples. Some of the tuples' entries can be guessed ("source node label", "target node label") others are hard to guess. In particular, the graph appears to be directed by construction, unless one defines all edges in a symmetric fashion.

3) Similarly, in Listing 1: conventions are not explained. Later it is mentioned that strings such as 'li_op/r_in' are defined in the YAML templates. At the moment some pieces of the listing hand in the air such that can only speculate about their meaning when reading the manuscript.

4) Comparatively, Listing 2 is better, but, for example, we have "method='DOP853'" in the listing, whereas in the text we have "a Runge-Kutta algorithm of order 8". This way of paraphrasing obscures, obscuring that the text is describing the listing (as it should). Similar instances occur for other listings.

5) Figure 3a cannot have been produced by the snippet shown in Listing 5. The reference also points to [6] (a standard review), not the underlying code reproducing Figure 3a in the repository.

Reviewer #3: Summary

======

The submission by Gast et al reports on the authors’ work developing a suite of Python-based software libraries aimed at simplifying and enhancing mathematical modelling research in neurobiology and other scientific fields. The principal library is PyRates (so-named in reference to its original intended scope of rate-based neuron and neural population models), an earlier version of which was described in a previous methods paper (Gast et al. PLoS One 2019). The present paper focuses on several major updates to the library since then, including: i) a [nominally] broader scope, up from from “rate-based neural models” to “dynamical systems in biology”; ii) a broader suite of code generation options and languages, and a greater emphasis on code generation per se; iii) two additional supporting libraries, PyCoBI (for parameter continuation) and RectiPy (for parameter optimization), that extend the functionality of PyRates considerably.

At a technical level, the principal innovation of PyRates is its approach to the ‘frontend’ problem of flexibly and efficiently specifying differential equation models via a hierarchical templating system, and their subsequent translation into ‘backend’ executable code in a range of programming languages. The template system consists of an ‘operator’ template and a ‘circuit’ template, that succeeds in capturing succinctly the essential structure of network models used in neuroscience and other fields. A particularly nice touch at the template parsing stage is the use of SymPy to extract symbolic graphs from the text-specified system equations., which then form the basis of the backend code generation process.

The currently supported backend implementation options are NumPy, TensorFlow, PyTorch, Fortran, Julia, and Matlab. Being able to switch easily between each of these implementation options has several clear advantages. For example, working with the NumPy and Matlab backends may be optimal for understanding, debugging, etc., as familiarity with these languages is more common than the others, in computational neuroscience at least. If execution speed is of priority, then the Fortran or Julia backends are likely to be preferable. If model optimization / parameter fitting is a requirement, then the PyTorch or TensorFlow backends may be preferred. This flexibility is an important feature of the library, although as noted below there is not much in the way of quantitative comparisons between these alternative backend options offered in the manuscript.

After a short technical description of the PyRates frontend and backend architectures and model definition interface, the paper goes on to present usage examples with corresponding code blocks for i) numerical simulations and parameter sweeps, ii) bifurcation analysis, and iii) parameter optimization, respectively using Van der Pol oscillator, quadratic integrate-and-fire, and delay-coupled leaky integrator models of neural dynamics. These and other examples are provided clearly and replicably on the software’s documentation website. From my own testing, I have found all of the code in the manuscript and the documentation pages to run without error and as expected.

Overall, the article is well written and compelling, both in its motivation and its example cases. The PyRates library does indeed provide a fairly unique alternative to many of the more established computational neuroscience, with several advantages that the authors describe clearly and succinctly. As is always the case, the most important factors for medium/long-term success of the project will be good documentation, community adoption, continual development, and (eventually) easily replicable high-profile published use cases. The present paper, along with its 2019 predecessor, provide a solid foundation for all of these.

I am therefore happy to recommend the paper for publication in PLoS CB, provided that the authors address the following points:

Major Points

========

Distinction from previous work

The authors’ previous methodological article on PyRates is mentioned three times in the manuscript:

Line 28: “PyRates was previously developed as a toolbox for neural network modelling [20]”.

Line 53: “We then present the software structure of PyRates in detail, noting novel features that have been added since our previous manuscript[20]”.

Line 130: “A detailed description of this template-based user interface is provided in our previous work [20], as well as in our online documentation”

Now, despite the presence of these lines, I am still a little concerned that the point of this article being an update on the previous one, and not a primary reference, is not made sufficiently clear. Given the amount of overlap in content with Gast et al. (2019), I think the authors need to do a little bit more to help communicate that this is the case, and what exactly motivates a follow-up article such as this one.

Specifically, for example,

- There is no mention of the authors’ previous work in the Abstract, Author Summary, or the entirety of the Discussion sections. Please fix this.

- Please adjust phrases like ‘we introduce’, for the above reasons.

- Where previous work is mentioned (eg line 29), please add some more sentences saying what was actually done previously, and how specifically that functionality is now being extended/augmented by the new library updates

Performance comparison of several backends

A key feature of the library is its ability to generate code in multiple scientific programming languages. One of the advantages of this would be to take advantage of the well known faster execution of some languages than others. However, the manuscript does not show any examples of this in action. It would be helpful to see some simple benchmark tests where the same model is run for all of the backend options, and run time is tracked. One would expect an order of something like Fortran, Julia, Numpy, PyTorch, Tensorflow, Matlab. The exact order and how much daylight there is between different languages is of course liable to vary considerably from application to application; although if there is a dramatic and consistent deviation from expectations here that would need to be clearly explained, as it would raise questions around the code generation approach. Secondary optimizations such as vectorization and jit compilation (as mentioned in the paper already) could be incorporated or left out, as the authors see fit.

Minor Points

========

Scope of library

-------------------

A notable shift in this paper relative to the 2019 one is its generalization of PyRates’ scope, from a tool accommodating a specific type of neural modelling, to one supporting dynamical systems modelling in general. All well and good, but it then comes as a bit of surprise that all of the examples shown continue to be of rate-base neural models. The broader and bolder scope that the authors are apparently aiming for would be a bit more convincing if they included examples from some other fields, such as population ecology or kinesiology.

Graphs

--------

Line 211 - The PyTorch graph is mentioned several times in this paragraph. Previously, the SymPy model graph has been mentioned. Please say a few words about the relationships between the two (e.g., SymPy graph is a subgraph of the PyTorch graph ?)

Run times

-------------

Line 267 states “To minimize the runtime of this problem, we use the function pyrates.grid_search, which takes a set of multiple model parameterizations and performs the numerical integration in a single combined model by vectorizing the model equations.”

Can the authors please provide a demonstration that this approach reduces the run time more effectively than comparable alternatives, eg multiprocessing/multithreading with Joblib?

Templates

------------

The paper would be a little more complete if a full model template is provided. Currently all of the examples shown use pre-defined templates, the contents of which are indicated only schematically in Figure 1A. As such the operator/circuit/template framework, which is a major feature of the PyRates USP, is not demonstrated as fully as it ideally would be.

Please could the authors add an additional figure / listing for one of the model examples that shows one of the templates in full.

Auto-07p

-----------

For the benefit of interested readers who are not familiar with the methodology, please provide a few further sentences describing how Auto-07p performs automated bifurcation analysis.

(suggest from around line 399 "Starting from this steady-state solution, we can perform parameter continuations and automated bifurcation analysis [6] via PyCoBi.")

Typographical / grammatical errors

Line 556 - 'see section '

(no section name/number given)

Line 552-554 -

Inconsistent switching between present and past tense

[ FYI: In terms of English language usage, this is one of the most tightly written and least error-strewn manuscripts I have ever reviewed. Good job. ]

**Have the authors made all data and (if applicable) computational code underlying the findings in their manuscript fully available?**

Reviewer #1: Yes

Reviewer #2: Yes

Reviewer #3: Yes

PLOS authors have the option to publish the peer review history of their article (what does this mean?). If published, this will include your full peer review and any attached files.

Reviewer #1: No

Reviewer #2: No

Reviewer #3: **Yes: **John D Griffiths
---

## [Decision Letter · Decision Letter 1]

14 Dec 2023

Dear Mr. Gast,

We are pleased to inform you that your manuscript 'PyRates - A Code-Generation Tool for Modeling Dynamical Systems in Biology and Beyond' has been provisionally accepted for publication in PLOS Computational Biology.

Best regards,

Daniele Marinazzo

Section Editor

PLOS Computational Biology

Daniele Marinazzo

Section Editor

PLOS Computational Biology

Reviewer's Responses to Questions

**Comments to the Authors: **

Reviewer #2: The revision addressed my main concerns. An interface for equation generation is perfectly sufficient for tools such as Auto-07p or DDE-Biftool, as this is a main barrier to entry for using the tools (apart from understanding the underlying maths). It should be noted, though, that these tools would benefit greatly from also providing derivatives of the function (similarly, optimization tools, I suppose). The library does not seem to provide derivatives automatically, which is a major omission, limiting the usefulness of the interface.

**Have the authors made all data and (if applicable) computational code underlying the findings in their manuscript fully available?**

Reviewer #2: Yes

PLOS authors have the option to publish the peer review history of their article (what does this mean?). If published, this will include your full peer review and any attached files.

Reviewer #2: No

---

## [Editor Report · Acceptance letter]

20 Dec 2023

PCOMPBIOL-D-23-01476R1 

PyRates - A Code-Generation Tool for Modeling Dynamical Systems in Biology and Beyond

Dear Dr Gast,

I am pleased to inform you that your manuscript has been formally accepted for publication in PLOS Computational Biology. Your manuscript is now with our production department and you will be notified of the publication date in due course.

With kind regards,

Anita Estes
